# Consensus Report on Preventive Antibiotic Therapy in Dental Implant Procedures: Summary of Recommendations from the Spanish Society of Implants

**DOI:** 10.3390/antibiotics11050655

**Published:** 2022-05-13

**Authors:** Angel-Orión Salgado-Peralvo, Alvaro Garcia-Sanchez, Naresh Kewalramani, Antonio Barone, Jose-María Martínez-González, Eugenio Velasco-Ortega, José López-López, Rodrigo Kaiser-Cifuentes, Fernando Guerra, Nuno Matos-Garrido, Jesús Moreno-Muñoz, Enrique Núñez-Márquez, Iván Ortiz-García, Álvaro Jiménez-Guerra, Loreto Monsalve-Guil

**Affiliations:** 1Department of Dental Clinical Specialties, Faculty of Dentistry, Complutense University of Madrid, 28040 Madrid, Spain; jmargo@odon.ucm.es; 2Science Committee for Antibiotic Research of Spanish Society of Implants (SEI—Sociedad Española de Implantes), 28020 Madrid, Spain; k93.naresh@gmail.com (N.K.); barosurg@gmail.com (A.B.); evelasco@us.es (E.V.-O.); 18575jll@gmail.com (J.L.-L.); dr.rkaiser@gmail.com (R.K.-C.); fguerra@ci.uc.pt (F.G.); nunogar@me.com (N.M.-G.); je5us@hotmail.com (J.M.-M.); enrique_aracena@hotmail.com (E.N.-M.); ivanortizgarcia1000@hotmail.com (I.O.-G.); lomonsalve@hotmail.com (L.M.-G.); 3Department of Oral Health and Diagnostic Sciences, School of Dental Medicine, University of Connecticut Health, Farmington, CT 06030, USA; ags.odon@gmail.com; 4Department of Nursery and Stomatology, Faculty of Dentistry, Rey Juan Carlos University, 28922 Madrid, Spain; 5Department of Surgical, Medical and Molecular Pathology and Critical Areas, Faculty of Dentistry, University of Pisa, 56126 Pisa, Italy; 6Department of Stomatology, Faculty of Dentistry, University of Seville, 41009 Seville, Spain; 7Department of Odontostomatology, Faculty of Dentistry, University of Barcelona, 08907 Barcelona, Spain; 8Faculty of Dentistry, Finis Terrae University, Santiago de Chile 7501015, Chile; 9Faculty of Dental Medicine, University of Coimbra, 3000-075 Coimbra, Portugal

**Keywords:** preventive antibiotic therapy, preventive antibiotics, antibiotics, oral implantology, dental implants, oral infections, dental implant failure, dental implant complications, statement

## Abstract

Current patterns of preventive antibiotic prescribing are encouraging the spread of antimicrobial resistance. Recently, the Spanish Society of Implants (SEI) developed the first clinical practice guidelines published to date, providing clear guidelines on how to prescribe responsible and informed preventive antibiotic therapy (PAT) based on the available scientific evidence on dental implant treatments (DIs). The present document aims to summarise and disseminate the recommendations established by this expert panel. These were based on the Preferred Reporting Items for Systematic Reviews and Meta-Analyses (PRISMA) statement. Studies were analysed using the Scottish Intercollegiate Guidelines Network (SIGN) checklist templates and ranked according to their level of evidence. They were then assigned a level of recommendation using the Grading of Recommendations, Assessment, Development and Evaluation system (GRADE). Guidelines were established on the type of PAT, antibiotic and dosage of administration in the placement of DIs without anatomical constraints, in bone augmentation with the placement of DIs in one or two stages, placement of immediate DIs, sinus elevations, implant prosthetic phase, as well as recommendations in patients allergic to penicillin. Therefore, the PAT must be adapted to the type of implant procedure to be performed.

## 1. Introduction

Dental implants (DIs) are the most predictable therapeutic option in the total or partial replacement of missing teeth; however, around 0.7–3.8% of them fail [1]. These failures can be “early” or “late”, depending on whether they occur before or after functional loading, respectively [2]. Early failure occurs as a result of a failure in osseointegration, derived from local and/or systemic factors, and represents 5% of all failures [3,4]. Since the beginning of Oral Implantology, the prescription of preventive antibiotics has been incorporated into DI placement protocols [5] due to the presence in the oral cavity of more than 500–700 bacterial species, in addition to other non-culturable microorganisms discovered by molecular biological techniques that can contribute to the development of postoperative infections [6,7]. Initially, this preventive prescription was generically referred to as “antibiotic prophylaxis”, but recently, the term “preventive antibiotic therapy” (PAT) has been defined to refer specifically to the preventive administration of antibiotics in healthy patients to avoid early DI failure and/or the development of postoperative infections [8].

Antibiotics are used for longer periods than other drugs in dentistry, such as anaesthetics, analgesics, anti-inflammatories or anxiolytics, among others, which increases the risk of adverse reactions, such as allergies that can cause life-threatening complications [9,10] or toxicity on various target organs, alterations in the usual microflora [11] and/or bacterial resistance. The latter occurs naturally, but the inappropriate and indiscriminate use of antimicrobials in humans, in food-producing animals and in the environment is accelerating the process. It is essential that the prescription and use of antibiotics are urgently changed because, even if new ones are developed, resistance will continue to pose a serious threat unless current prescribing patterns are changed [12].

The current evidence is very limited. Despite this, it has been shown that for every 24–50 healthy patients treated with PAT, early failure will be avoided in one patient [13,14,15,16] and only 1 in 143 will avoid postoperative infection [17]. The value of this risk reduction must be placed in the context of the emerging problems with antibiotic resistance before robust guidelines can be formulated, and the biological cost of DI failure must be weighed against the economic cost incurred, as the fear of infection and fear of legal and economic repercussions motivate the vast majority of prescriptions of these drugs [18].

Antimicrobial resistance causes more than 33,000 deaths per year in the European Union (EU) [19] and the associated healthcare costs and lost productivity are estimated at EUR 1.5 billion per year, which, extrapolated to national figures, represents a cost of around EUR 150 million per year [20]. According to data from the Minimum Basic Data Set Registry [21] (MBDS), in 2016, 2956 people died in Spain as a result of this type of infection. If urgent action is not taken, in 35 years, the number of deaths attributable to multidrug-resistant infections will reach 390,000 deaths per year across the EU (around 40,000 deaths per year in Spain) and resistance will overtake cancer as the leading cause of death [20].

Recent surveys [22,23,24] investigated patterns of PAT prescription in various procedures related to DI placement, revealing that perioperative prescription, i.e., before and after surgery, is the most frequent, followed by postoperative prescription only, which implicitly implies the administration of a higher amount of antibiotics than if only a preoperative regimen was applied. These data are alarming in light of previously reported data on antimicrobial resistance. To this end, the Spanish Society of Implants (SEI—Sociedad Española de Implantes) developed the first clinical practice guidelines [25] published to date, providing clear guidelines on how to perform responsible and informed PAT prescribing, based on the available scientific evidence (Appendix A). The present document aims to summarise and disseminate the recommendations established by this expert panel.

## 2. Materials and Methods

For the elaboration of the consensus report document, a group of 12 expert dentists and stomatologists was carefully selected for their extensive knowledge on the subject. It also benefited from the assessment of three external reviewers who participated as independent evaluators and the validation of the following Scientific Societies: Spanish Society of Implants (SEI), Spanish Society of Gerodontology (SEGER), Portuguese Society of Implantology and Osseointegration (SOPIO), Latin American Oral Implantology Society (SIOLA) and the International Federation of Oral Implantology (FIIO), which includes the following scientific societies: Colombian Dental Implant Association (SOCI), Mexican College of Oral and Maxillofacial Implantology (CMIBM), Chilean Society of Oral and Maxillofacial Implantology, Brazilian Academy of Osseointegration (AMBROSS) and Peruvian Association of Integral Oral Implantology (ASPIOI).

This consensus report was structured according to the Preferred Reporting Items for Systematic Reviews and Meta-Analyses (PRISMA) statement [26].

### 2.1. Focused Questions

The study aimed to answer the following PICO (P = patient/problem/population; I = intervention; C = comparison; O = outcome) questions based on the PRISMA guidelines (Table 1, Table 2, Table 3, Table 4, Table 5 and Table 6):

The secondary objective was to determine the type of PAT, type of antibiotic, dose and posology recommended in these cases according to the available scientific evidence.

### 2.2. Evaluation and Evidence Synthesis

The studies were evaluated by two authors, determining the quality of the scientific evidence using the Scottish Intercollegiate Guidelines Network (SIGN) [27] checklist templates. Disagreements between the two were solved by the intervention of a third author. The level of evidence was classified by the following grading:

1++: High-quality meta-analyses, systematic reviews of clinical trials (CTs) or high-quality CTs.

1+: Well-conducted meta-analyses, meta-analyses of CTs or well-conducted CTs with low risk of bias.

1-: Meta-analyses, systematic reviews of CTs or CTs with a high risk of bias.

2++: High-quality systematic reviews of cohort or case-control studies. Cohort or case-control studies with a very low risk of bias and a high probability of establishing a causal relationship.

2+: Well-conducted cohort or case-control studies with low risk of bias and with a moderate probability of establishing a causal relationship.

2-: Cohort or case-control studies with a high risk of bias and a significant risk that the relationship is not causal.

3: Non-analytical studies, such as case reports and case series.

4: Expert opinion.

After having catalogued each of the pieces of evidence that respond to the initial (PICO) question, the recommendation is made by incorporating a Grade of Recommendation based on the GRADE [28] (The Grading of Recommendations Assessment, Development and Evaluation) approach:

A: at least one meta-analysis, systematic review or CT 1++; or a volume of evidence composed of studies classified as 1+ and with high consistency between them.

B: evidence comprised of studies classified as 2++ directly applicable to the guideline’s target population, with high consistency between them; or clinical evidence extrapolated from studies classified as 1++ or 1+.

C: scientific evidence formed by studies classified as 2+ directly applicable to the guideline’s target population and demonstrating high consistency between them; or scientific evidence extrapolated from studies classified as 2++.

D: level 3 or 4 scientific evidence, or evidence extrapolated from studies classified as 2+.

### 2.3. Clinical Relevance

To the best of our knowledge, this is the first consensus report on PAT in DI procedures published to date. Due to the massive number of DI treatments performed worldwide, it has been considered essential to establish clear guidelines in this regard, so that more responsible and effective use of these drugs can be carried out.

### 2.4. Information Sources and Search Strategy

The MEDLINE database (via PubMed) was used as the search engine for the available evidence. Its elaboration was carried out from January 2020 to February 2021. The reference list of the included studies was also reviewed for possible inclusion.

A preliminary search was conducted to define the scope and objectives of the consensus report. This search determined the eligibility criteria of the studies to be included given the available evidence for each procedure related to the placement of DIs. The objective was to gather the studies with the best evidence. If there was a limited amount of scientific literature for a given treatment, we moved down the pyramid of evidence and, if no related studies were found, we evaluated those carried out on teeth whose results could be extrapolated. Thus, the search was individualised for each DI treatment as follows:

#### 2.4.1. PICO Question 1

##### PubMed Strategy

(dental implant OR dental implants OR dental implantology OR oral implantology) AND (antibiotics OR preventive antibiotics OR antibiotic prophylaxis).

##### Filters Applied

(a) human studies; (b) meta-analyses and systematic reviews; (c) articles published in English and/or Spanish, and (d) from 2010 to 2020.

#### 2.4.2. PICO Question 2

##### PubMed Strategy

(immediate implant OR immediate implantation OR fresh extraction socket) AND (dental implant OR dental implants OR dental implantology OR oral implantology) AND (antibiotics OR antibiotic prophylaxis OR clindamycin OR amoxicillin OR azithromycin OR erythromycin).

##### Filters Applied

(a) human studies; (b) meta-analyses and systematic reviews; (c) articles published in English and/or Spanish, and (d) from 2010 to 2020.

#### 2.4.3. PICO Question 3

##### PubMed Strategy

(maxillary sinus lift OR maxillary sinus augmentation OR sinus lift elevation) AND (antibiotics OR antibiotic prophylaxis OR clindamycin OR amoxicillin OR erythromycin OR azithromycin OR metronidazole).

##### Filters Applied

(a) human studies; (b) no filters by study type; (c) articles published in English and/or Spanish. (d) No temporal restrictions were applied and (e) the search was updated to December 2020.

#### 2.4.4. PICO Question 4

##### PubMed Strategy

(bone grafting OR alveolar ridge augmentation OR alveolar bone grafting OR bone graft augmentation OR guided bone regeneration OR bone block) AND (dental implants OR dental implant OR oral implantology OR dental implantology) AND (antibiotic prophylaxis OR antibiotics).

##### Filters Applied

(a) human studies; (b) meta-analyses, systematic reviews and randomized clinical trials (RCTs); (c) articles published in English and/or Spanish, and (d) from 2005 to 2020.

#### 2.4.5. PICO Question 5

##### PubMed Strategy

(peri-implant plastic surgery OR periodontal plastic surgery OR free gingival graft OR connective tissue graft OR graft OR second stage surgery OR prosthetic phase OR implant-supported prosthesis) AND (antibiotics OR antibiotic prophylaxis) AND (dental OR dental implant OR dental implants OR oral implantology OR dental implantology).

##### Filters Applied

(a) human studies; (b) meta-analyses, systematic reviews, RCTs, observational studies, multicentre studies, comparative studies, and clinical studies; (c) articles published in English and/or Spanish, and (d) from 2000 to 2020.

#### 2.4.6. PICO Question 6

##### PubMed Strategy

(penicillin allergy OR clindamycin) AND (dental implant OR dental implant failure).

##### Filters Applied

(a) human studies; (b) meta-analyses, systematic reviews, RCTs, observational studies, multicentre studies, comparative studies, and cohort studies; (c) articles published in English and/or Spanish. (d) The search was not temporarily restricted and was updated to December 2020.

Data collection was conducted using a predetermined table designed in advance of the assessment of the resulting articles.

## 3. Results

### 3.1. Study Selection

For the evaluation of the scientific evidence, the articles were reviewed by two authors using the SIGN critical reading template. Disagreements between the two were resolved by the intervention of a third author.

The recommendations for each of the questions asked are summarised below, providing recommendations on the type of antibiotic and dosage when possible:

### 3.2. PAT Prescribing Guidelines for DI Placement in Routine Situations in Healthy Patients

A large number of systematic reviews and/or meta-analyses on the effects of PAT administration on early DI failure and/or postoperative infections have been published in the last 10 years. Specifically, 11 studies were found that answered the established PICO question [13,14,15,16,17,29,30,31,32,33,34], of which 2 were classified with a level of evidence (LoE) 1- [15,33], 8 with a 1+ [13,16,17,29,30,31,32,34], and 1 with a 1++ [14].

In general, there is great heterogeneity among the different RCTs on which these investigations are based, as most of them used oral amoxicillin, except for some authors who used other types [13,14]. In addition, the amoxicillin regimens and doses varied widely. All protocols were effective in reducing early DI failure compared to no PAT prescription or placebo (odds ratio (OR) _mean_ = 0.08–0.45). More specifically, a significant benefit has been demonstrated in the use of preoperative antibiotics [17,31] (LoE 1+). No additional benefit is observed when combining amoxicillin with clavulanic acid [17,29] or amoxicillin postoperatively or perioperatively (LoE 1+); instead, they increase the risk of adverse reactions as the regimens are extended over a longer period. Despite this, patients treated with PAT have only a 1.8% higher risk than those not treated [30] (LoE 1+).

Their influence on the prevention of postoperative infections was also evaluated in six studies [13,15,16,17,30,32], of which four provided data globally and two specifically on how pre-and/or postoperative administration affects infection rates [17,32]. The various studies were unanimous in determining that, in healthy patients, there is no significant difference between not prescribing PAT or prescribing a placebo compared to not prescribing antibiotics [13,15,16,17,30,32] in the risk of developing early (1–2 weeks) and/or late (3–4 months) infections [32]. The mean number needed to treat (NNT) (i.e., the number of individuals who must be treated to prevent an adverse event compared to the expected outcomes in the control group) for avoiding postoperative infection is 143. Specifically, the NNT for preoperative amoxicillin prescriptions is 100 and for postoperative prescriptions 143 [17].

In short, there is a tendency to recommend the routine prescription of PAT in these cases; however, a smaller number of authors consider that its use should be avoided in simple cases in healthy patients [14,30,31,33] (LoE 1- [*n* = 1], 1+ [*n* = 2]), 1++ [*n* = 1]). These authors base their conclusions on the fact that PAT offers a modest reduction in early DI failure of 1.8–4% [14] (LoE 1++).

Only three authors studied the recommended guidelines in these cases [16,17,29]. Rodríguez-Sánchez et al. [17], based on the recommendations of the Cochrane Collaboration [16] (2013), concluded that only preoperative PAT with amoxicillin at a dose of 2 or 3 g 1 h preoperatively is effective (LoE 1+). A year later, Romandini et al. [29] conducted a network meta-analysis—which allows more than two interventions to be compared simultaneously as the only better alternative would be to conduct an RCT with several thousand participants, which is quite complex—concluding that the most effective protocol in preventing DI failures is the administration of 3 g of amoxicillin one hour before (OR = 0.41). The most studied protocol (2 g 1 h before) has only a 0.2% chance of being the best (LoE 1+).

#### PICO Question 1

PAT reduces the rate of early DI failure in healthy patients, but not the risk of infection. Postoperative or perioperative regimens are not justified as they have not shown additional benefits to preoperative prescription and increase the likelihood of adverse drug reactions. Therefore, it is recommended to prescribe PAT preoperatively, specifically 2 to 3 g of amoxicillin 1 h before surgery (GRADE A). However, not prescribing it could not be considered a wrong approach in certain cases (GRADE B).

### 3.3. PAT Prescribing Guidelines in Healthy Patients for Immediate DI Placement

After evaluation of the selected articles, six articles with a level of evidence of 2++ were included. All were systematic reviews [35,36,37,38,39] and one of them was also a meta-analysis [40].

Cosyn et al. [40] reported immediate DI failure rates of 5.1% compared to 1.1% for delayed placement, i.e., 6 months after extraction (relative risk (RR) = 0.96, *p* = 0.02), with all failures being early failures. A trend toward lower survival of immediate DIs was observed when PAT was not administered postoperatively (RR = 0.93). On the other hand, in both DI placement protocols (immediate vs. delayed), healing was adequate, except in one study [41], where they found 5-times higher risk of surgical wound complications in immediate DIs (26.1% vs. 5.3%, respectively) (LoE 2++).

Lee et al. [39] concluded that there is no specific protocol for antibiotic regimen in these treatments but acknowledged the need to prescribe them (LoE 2++).

Chrcanovic et al. [35] conducted a systematic review, including studies that investigated the prognosis of immediate DIs in infected beds. They included animal (*n* = 7) and human (*n* = 21) studies, none of which compared immediate DI placement with and without the prescription of PAT, so there is no control group with which to compare results. If only human studies are considered and all cases of immediate DIs are included, without distinction between the previous pathology or not, the failure rate is 1.7%. The total duration of PAT in the different studies was 6–14 days. The most frequent regimen was perioperative, although some studies carried out only pre-or postoperative PAT (LoE 2++).

Álvarez-Camino et al. [38] highlighted the need to prescribe PAT in immediate DIs in infected beds; however, they did not recommend a specific guideline (LoE 2++).

Lang et al. [37] conducted a systematic review, including 46 studies, of which 33 prescribed PAT: 4 carried out preoperative PAT (*n* = 244 DIs), and in 15, only postoperative PAT, which lasted 5–7 days (*n* = 935 DIs). The remaining 14 studies prescribed perioperative PAT (one preoperative dose followed by 5–7 days postoperatively) (*n* = 665 DIs). To determine the DI failure rates associated with each regimen, they performed a multivariate analysis using the fixed-effect Poisson regression model, taking the preoperative prescription as a reference. In this way, they calculated the annual failure rate of DIs placed under preoperative PAT at 1.9%, postoperative at 0.5% and perioperative at 0.8% (*p* = 0.002). Therefore, single-dose preoperative PAT alone is not sufficient to maintain bacterial levels below the critical threshold during the healing period but prescribing them 5–7 days post-surgery may help to prevent complications that may lead to DI failure (LoE 2++).

Waasdorp et al. [36] did not elaborate on the recommended antibiotic regimen but, despite stating that there is controversy about its use, they recommend prescribing PAT for immediate DIs in infected sites. The regimens were very heterogeneous, with durations ranging up to 31 days. Failure rates ranged from 0 to 8%. In studies where they were prescribed postoperatively, the failure rate was 0–2.3% (*n* = 4), in those prescribed preoperatively 8% (*n* = 1) and perioperatively 0–2.6% (*n* = 2) (LoE 2++).

The type of PAT and the guidelines used were not provided by all studies. From those that did provide these data [35,36,38], it can be extracted that they were very heterogeneous, reflecting a lack of consensus.

#### PICO Question 2

Evidence has shown an added benefit of perioperative PAT prescription in reducing early failure of immediate DIs. Despite this, studies have failed to recommend a specific type and dose of PAT and, therefore, pending further research, it is necessary to advise a specific guideline based on the extrapolation of the recommendations established in Endodontics, given the nature of the microbiota to be combated, advocating for the prescription of PAT with a loading dose, followed by a maintenance dose (GRADE D). It is recommended to administer the loading dose of 2 or 3 g of amoxicillin 1 h before surgery (GRADE B), followed by 500 mg/8h, during the 5–7 postoperative days (GRADE D). In case of confirmed true penicillin allergy, azithromycin 500 mg 1 h before followed by 250 mg/24 h, 5–7 days; clarithromycin 500 mg 1 h before followed by 250 mg/12 h, 5–7 days; or metronidazole 1 g 1 h preoperatively followed by 500 mg/6 h/5–7 days (GRADE D).

### 3.4. PAT Prescribing Guidelines for Healthy Patients Undergoing Sinus Lifts with Single or Two-Stage DI Placement

The evidence related to PAT in sinus lifts is very limited. After searching, one study was found [42] that examined whether PAT prevents failure of DIs placed in these procedures (LoE 2+). Recommendations on how to prescribe them in these cases could be extracted from two studies (LoE 2+ [43] and 4 [44]). No studies were found that provided information on the effect of these drugs on the prevention of postoperative infections.

Zinser et al. [42] evaluated risk factors in sinus lifts with 1- or 2- stage DI placement, showing that prescribing PAT does not significantly influence DI or graft failure rates compared to not prescribing PAT (LoE 2+). Some authors only recommend prescribing PAT if Schneider’s membrane perforation occurs because of the high failure rate due to graft infection [45] (LoE 2+) (OR = 16.82 [46]). The rate of sinus membrane perforation in these procedures has been estimated at 18.3% [46]–23.5% [47]. Of these, 11.3% will experience sinusitis and infection (compared to 1.4% in the case of no perforation), most likely due to colonisation by native sinus bacteria. These authors recommend applying, in the case of non-perforation, the same protocols applied to the placement of ordinary DIs [45], i.e., 2 or 3 g of amoxicillin 1 h before surgery [29,48] (LoE 1+).

Carreño-Carreño et al. [43] took microbiological samples from subsinusal cavities during 227 sinus lifts with a lateral window approach. They did not prescribe PAT or chlorhexidine preoperatively and, in this study, amoxicillin 1 g/12 h was administered postoperatively. Once the results were obtained at 48 h, 81.9% of the patients were interrupted on antibiotic treatment because no bacterial species were found, i.e., they presented sterile sinuses at the time of sampling, while in the remaining 18.1%, antibiotic treatment was continued when a positive culture was obtained. In the case of a positive culture, the duration of antibiotic treatment was not specified (LoE 2+).

The choice of the type of PAT should be made based on the sensitivity of the micro-organisms following an antibiogram. In the case of empirical prescription of antibiotics, which is done in routine clinical practice, amoxicillin/clavulanic acid, ampicillin or ciprofloxacin are recommended as the first choice, since the germs found are sensitive to these antimicrobials, while they have shown greater resistance to macrolides, fosfomycin or penicillin G [43] (LoE 2+). Cephalosporins have shown modest efficacy and the use of clindamycin has been associated with an increased risk of graft failure (6%) compared to the group prescribed amoxicillin (0%) [49] (LoE 3).

An expert panel recommended amoxicillin/clavulanic acid 875/125 mg every 12 h, starting one day before surgery, followed by the same schedule, every 8 h for 7 days. In patients allergic to penicillin, clarithromycin 250 g/12 h together with metronidazole 500 mg/8 h, starting one day before surgery, followed by the same regimen for 7 days [44] is advised (LoE 4); however, the use of macrolides is not justified in these cases, so ciprofloxacin [43] is recommended (LoE 2+). The recommended doses of this drug for preventive use have not been described; however, in the treatment of chronic sinusitis, its efficacy has been demonstrated at a dose of 500 mg/12 h/9 days [49] (LoE 2++).

Khoury et al. [49] (LoE 3) used, in self-declared penicillin-allergic patients (i.e., not diagnosed by specific tests), clindamycin 600 mg 1 h preoperatively followed by 300 mg/ 8h/ 7 days, while in the non-allergic group, they prescribed amoxicillin 2 g preoperatively followed by 10 days postoperatively, in sinus lifts with lateral window approach with one or two-stage DI placement. Graft infection occurred in 0.48%, all of whom were “allergic” to penicillin, which accounted for 6% of these patients. Symptomatology started at 4–8 weeks. None of the patients had a history of sinusitis and there were no surgical complications, such as sinus membrane perforation, mucosal dehiscence, graft exposure and/or tissue necrosis.

The use of topical antibiotics, such as metronidazole, has also been described. For this, 5 mL of a sterile 0.5% metronidazole solution (25 mg) is applied as follows: 3 mL to irrigate the sinus after membrane elevation and 2 mL to hydrate the graft, which is equivalent to 1/20 of a 200 mg tablet, reducing the possibility of antimicrobial resistance. This results in a significant decrease in the number of inhomogeneous areas of the graft over the next three months, leading to a more compact and higher quality graft. These air bubble gaps suggest an anaerobic bacterial activity that increases the risk of graft failure (“septic theory”) [50] (LoE 2+).

#### PICO Question 3

The evidence suggests that the prescription of PAT does not reduce the failure rate of DIs placed at the same time as sinus lifts compared to not prescribing it. There is no evidence of their effect on the prevention of postoperative infections. Assuming that PAT does prevent them, their prescription should be based on prior cultivation (GRADE C); however, it is an impractical approach. Prescription of 2 to 3 g of amoxicillin 1 h before surgery would be sufficient in the absence of Schneider’s membrane perforation (GRADE B). In most cases, it is not possible to foresee this complication, so it is recommended to rely on the assumption that such a complication would occur. In this case, amoxicillin/clavulanic acid 875/125 mg/12 h, starting one day before surgery, followed by the same regimen, every 8 h for 7 days (GRADE D) is advised. In penicillin-allergic patients, ciprofloxacin 500 mg/12 h/9 days is recommended (GRADE D).

### 3.5. PAT Prescribing Guidelines in Healthy Patients in GBR, with Single or Two-Stage DI Placement

Overall, few studies investigated the effect of PAT on the prevention of postoperative infections after GBR, with or without simultaneous DI placement, and early DI failure. After evaluation of the selected articles, one systematic review [51] and four RCTs were included [52,53,54,55].

The systematic review [51], rated with an LoE 1++, concluded that prescribing PAT improves the rate of postoperative infections; however, they could not clarify whether a single dose is sufficient or whether it is necessary to prolong its administration beyond the day of surgery.

Of the four RCTs included, three employed a preoperative antibiotic dose in both the test and control groups. The first [52] (LoE 1+) studied treatment with 600 mg clindamycin 1 h before surgery and, in the test group, in addition, 300 mg/6h/1 day postoperatively vs. placebo, in bone augmentations with bone blocks covered with barrier membranes. The second RCT [53] (LoE 1+) compared the effect of a preoperative dose of 2 g fenetylline or 600 mg clindamycin in bone blocks covered by barrier membranes. Both studies showed lower failure rates in the group prescribed a single preoperative dose of clindamycin, but without significant differences. The RCT by Lee et al. [54] (LoE 1-) studied the effect of 2 g of a first-generation cephalosporin. Post-surgery, they prescribed 1 g/8 h/3 days vs. placebo in the test group, with no significant differences. Finally, the multicentre RCT led by Payer et al. [55] (LoE 1++) was the only one to compare perioperative PAT administration (2 g amoxicillin 1 h before surgery, followed by 500 mg/8 h/3 days) vs. placebo, with no significant differences. However, at the clinical level, suppuration was higher in the control group. Failure rates of DIs placed in one stage were lower in the control group compared to the test group (97.4% vs. 99.2%; *p* > 0.05). The authors concluded that there is no evidence to recommend the routine prescription of PAT in these interventions. Infection of the grafted material leads to its total loss [52,53] or partial loss (in the case of mucosal opening at 7–8 weeks post-surgery without clinical signs of infection) [53] and, it is suggested that, in the case of placing the DIs in one stage, it could be a risk factor for failure of osseointegration, as it could lead to an increase in the local inflammatory response [56,57].

#### PICO Question 4

Different studies have shown a reduction in the rate of postoperative infections in those cases where PAT was prescribed preoperatively compared to perioperatively (GRADE A). A single dose of 2 or 3 g amoxicillin 1 h before surgery is recommended to reduce the failure rate of single-stage DIs and to reduce the degree of bacterial contamination of the grafted bone particles in these cases, as well as in two-stage DIs (GRADE C).

### 3.6. PAT Prescribing Guidelines for Healthy Patients in the Prosthetic Phase of DIs (Second Stages, Impression Taking and Placement of the DI-Supported Prosthesis

The search did not find any articles investigating the appropriateness of prescribing PAT for the second stage of DIs, so the information obtained from four studies on surgical procedures in periodontology was extrapolated. Of these, three [58,59,60] had an LoE 2+, and one had an LoE 4 [61]. No studies on PAT in DI impression-taking and/or prosthetic placement were found.

Liu et al. [58] conducted a systematic review of RCTs (LoE 2+), in which they analysed the effect of PAT in periodontal access surgeries and/or regenerative periodontal surgeries, excluding mucogingival surgeries. The postoperative infection rate was very low, both in the test group (PAT), 0.073%, and in the control (no prescription of PAT), 0.693% (*p* < 0.05). However, only 0.170% of all surgeries experienced infectious complications and the NNT to avoid postoperative infection is 203, so the benefit of using PAT in these cases is considered to be of no clinical significance.

Oswal et al. [59] performed an RCT (LoE 2+), in which they analysed the effect of 1 g amoxicillin 1 h preoperatively vs. amoxicillin 500 mg/8h/5 days postoperatively and vs. not prescribing PAT in periodontal access surgery, mucogingival surgery, periodontal regeneration, osteoplasty and crown lengthening. No postoperative infections were reported, and they recommend not to prescribe these drugs in healthy patients, except in long surgeries (>2 h duration) or when biomaterials are grafted extensively. Powell et al. [60] studied the influence of PAT in procedures related to periodontal flap elevation. They observed that when soft tissue grafts are incorporated, the infection rate is 4%, compared to 1.9% when they are not used. Specifically, the infection rate after connective tissue grafts is 3.7% and after free gingival grafts 5.9%. However, the infection rate is lower when these drugs are not administered compared to when they are used pre- and/or postoperatively (1.8% vs. 2.9%), although without significant differences (LoE 2+).

An expert panel, the 10th European Workshop on Periodontology [61], concluded that systemic prescription of peri- or postoperative PAT is not indicated in periodontal plastic surgeries, although in extensive surgeries, local or systemic PAT might be indicated (LoE 4).

#### PICO Question 5

At present, PAT in the second DI stages, impression making and/or DI prosthetic placement does not seem to be justified (GRADE D).

### 3.7. Prescription of Antibiotics Other than Amoxicillin for DI Placement in Healthy Patients

Five studies were found that answered the PICO question posed. All were observational and, in particular, four were cohort studies [62,63,64,65] (LoE 2+) and one case series [49] (LoE 3).

The included studies only evaluated the effect of clindamycin as an alternative to amoxicillin in the placement of DIs in native bone, with or without the need for simultaneous GBR and/or sinus lifts and immediate DIs [49,62,63,64,65]. Of the five studies, allergy testing to confirm the diagnosis was performed in only one study [65]. The remaining studies included patients with self-reported allergies. Additionally, an investigation [66] evaluating the effect of azithromycin vs. amoxicillin was included (LoE 2++).

Salomo-Coll et al. [62] (LoE 2+) described failure rates in non-allergic patients as 4-times lower (RR = 3.84) than in allergic patients (8% vs. 24.7%). In allergic patients, 21.1% of DIs failed late, while 79% failed early, as a consequence of a failure in the osseointegration process (80%) or uncontrolled infections (20%). At the patient level, failure rates were 5.2% in non-allergic and 18.9% in allergic patients (*p* = 0.046) (RR = 3.64) [62].

French et al. [65] (LoE 2+) found twice the risk of DI failure in allergic patients treated with clindamycin vs. those treated with amoxicillin (hazard ratio (HR) = 2.16); however, these results were not significant due to the low number of allergic patients included and the low failure rates experienced in the whole sample (0.7%). These authors suggest avoiding immediate DI placement if penicillin cannot be administered. The same working group, two years later, published a similar study [63] (LoE 2+), in which they described DI failure rates in non-allergic patients of 0.8% (of these, 53.8% were early failures) vs. 2.1% in allergic patients (80% failed early) (*p* = 0.002), with an OR of 3.10. They also investigated the occurrence of postoperative infections, which was 0.6% in non-allergic and 3.4% in allergic patients, i.e., 6-times higher. Further, 12.3% of the DIs were immediate (*n* = 687), of which 91.7% (*n* = 630) were placed in non-allergic patients, with failure rates of 1%, while 8.3% in allergic patients, with failure rates of 10.5%, i.e., 10-times higher. The differences were due to a higher infection rate in “allergy sufferers”.

Wagenberg and Froum [64] conducted a similar investigation, in which they described a 5.7-fold increased risk of immediate DI failure secondary to infection in allergy sufferers prescribed clindamycin (8.5%) compared to non-allergy sufferers given amoxicillin (3%; RR = 3.34), with significant differences (LoE 2+).

Khoury et al. [49] (LoE 3) administered clindamycin 600 mg 1 h preoperatively followed by 300 mg/8 h/7 days postoperatively in “allergy sufferers”, while in non-allergic patients, they administered amoxicillin 2 g preoperatively followed by 10 days postoperatively, in sinus lifts with a lateral window approach, with one- or two-stage DI placement. Subantral graft infection occurred in 0.5%, all in “allergy sufferers”, which accounted for 6% of these patients. The infection occurred in the subantral graft and the symptomatology started at 4–8 weeks. None of the patients had a history of sinusitis and no surgical complications such as sinus membrane perforation, mucosal dehiscence, graft exposure and/or tissue necrosis occurred.

A study [66] (LoE 2++) evaluated azithromycin 500 mg compared to amoxicillin 2 g, both 1 h before DI surgery. On day 6, they found concentrations of 3.4 (0.7) and 2.8 g/mL (0.9) in gingival and peri-implant crevicular fluid, respectively, while amoxicillin concentrations were below detectable limits. Furthermore, gingival crevicular fluid levels were significantly lower in the azithromycin group during the initial healing period. Therefore, azithromycin acts on inflammation and early healing by decreasing levels of granulocyte colony-stimulating factor (G-CSF), interleukins 6 and 8, macrophage inflammatory protein 1 (MIP-1) and interferon-induced 10 kDa protein (IP-10), reducing mobilisation of granulocyte precursors and recruitment of immune and inflammatory cells during the healing phase. In addition, its bioavailability is higher compared to amoxicillin and clindamycin.

#### PICO Question 6

The prescription of clindamycin has a significantly elevated risk of DI failure related to osseointegration failure and a risk of infection up to 6-times higher than in patients prescribed amoxicillin. In turn, immediate DIs have an increased risk of failure in these cases (GRADE C). Pending further studies, azithromycin 500 mg 1 h before surgery is recommended (GRADE C).

## 4. Discussion

Dentists are often faced with the dilemma of whether or not to prescribe antibiotics preventively in DI treatment, which is currently a controversial issue. Prescription has been accepted to avoid systemic bacteraemia [67], but also to achieve an adequate antibiotic concentration in the blood to prevent bacterial contamination during DI surgery or grafting [54], since the oral cavity, per se, is a septic cavity. Despite this, the routine prescription of PAT in healthy patients does not present a justified risk–benefit ratio [13,14,68].

These recommendations (Table 7) are intended to raise awareness and encourage responsible antibiotic use among professionals who perform DI treatments because, if new antibiotics are developed and current prescribing patterns remain unchanged, resistance will continue to pose a serious threat.

To date, the largest number of publications and the highest level of scientific evidence (systematic reviews and/or meta-analyses) refer to the prescription of PAT in healthy patients without anatomical conditions. Recent advances have been made in establishing recommendations for other DI procedures, such as bone augmentation with single- or two-stage DI placement [48], or the placement of immediate DIs [69]. However, this evidence is limited and there are several types of DI treatments for which no clear recommendations have yet been established. This is essential as most professionals have a more clinical profile, and to achieve informed prescribing, it is a priority to provide them with clear guidelines.

### Strengths and Limitations

This consensus report presents several strengths, such as the searching process of studies and data extraction performed in duplicate. Furthermore, these recommendations were endorsed by several international societies, such as SEGER, SIOLA, SOPIO and FIIO (with their respective member societies). In addition, on 12 December 2021, they were endorsed by the General Council of Dentists of Spain.

Nonetheless, these recommendations have limitations, such as the shortage of studies with control groups that allow a comparison between the groups, which is why the external validity of the results in this review should be confirmed with future studies.

## 5. Conclusions

PAT is indicated for all types of implant surgical procedures, except for the prosthetic phase of DIs. The type of PAT, and its dosage, must be adapted to the type of implant procedure to be performed. Furthermore, the prescription of PAT shall in no way replace or relax the adoption of the necessary aseptic and sterile measures in surgical procedures.

## Figures and Tables

**Table 1 antibiotics-11-00655-t001:** PICO question 1.

Clinical Problem	PAT ^1^ Prescribing Guidelines for DI ^2^ Placement in Routine Situations in Healthy Patients.
Population (P)	Healthy patients to be treated with DIs without simultaneous GBR ^3^.
Intervention (I)	Prescription of PAT.
Comparison (C)	No prescription of PAT.
Outcome (O)	DI failure and postoperative infection.
PICO question	In healthy patients to be treated with DIs without the need for simultaneous GBR (P), does the prescription of PAT (I), compared with no prescription of PAT (C), modify the rate of DI failure and/or postoperative infection (O)?

^1^ PAT—preventive antibiotic therapy; ^2^ DI(s)—dental implant(s); ^3^ GBR—guided bone regeneration.

**Table 2 antibiotics-11-00655-t002:** PICO question 2.

Clinical Problem	PAT ^1^ Prescribing Guidelines in Healthy Patients for Immediate DI ^2^ Placement.
Population (P)	Healthy patients undergoing immediate DI placement, with or without the presence of chronic infection of the tooth to be extracted.
Intervention (I)	Prescription of PAT.
Comparison (C)	No prescription of PAT.
Outcome (O)	DI failure and postoperative infection.
PICO question	In healthy patients to be treated with immediate DIs, with or without infection of the tooth to be extracted (P), does PAT prescription (I), compared with no PAT prescription (C), modify the rate of DI failure and/or postoperative infection (O)?

^1^ PAT—preventive antibiotic therapy; ^2^ DI(s)—dental implant(s).

**Table 3 antibiotics-11-00655-t003:** PICO question 3.

Clinical Problem	PAT ^1^ Prescribing Guidelines for Healthy Patients Undergoing Sinus Lifts with Single or Two-Stage DI ^2^ Placement.
Population (P)	Healthy patients treated with sinus lifts with a lateral window or transcrestal approach, with one or two-stage placement of DIs.
Intervention (I)	Prescription of PAT.
Comparison (C)	No prescription of PAT.
Outcome (O)	Failure of DIs placed in one stage and postoperative infection.
PICO question	In healthy patients to be treated with sinus lifts through a lateral window approach or transcrestal approach, with single or two-stage DI placement (P), does the prescription of PAT (I), compared to not prescribing PAT (C), modify the rate of DI and/or graft failure (O)?

^1^ PAT—preventive antibiotic therapy; ^2^ DI(s)—dental implant(s).

**Table 4 antibiotics-11-00655-t004:** PICO question 4.

Clinical Problem	PAT ^1^ Prescribing Guidelines in Healthy Patients in GBR ^2^, with Single or Two-Stage DI ^3^ Placement.
Population (P)	Healthy patients to be treated with GBR procedures, with or without simultaneous placement of DIs.
Intervention (I)	Prescription of PAT.
Comparison (C)	No prescription of PAT.
Outcome (O)	Failure of DIs placed in one stage and postoperative infection.
PICO question	In healthy patients to be treated with GBR procedures, with or without simultaneous placement of DIs (P), does the prescription of PAT (I), compared to the non-prescription of PAT (C), modify the rate of DI and/or postoperative infections (O)?

^1^ PAT—preventive antibiotic therapy; ^2^ GBR—guided bone regeneration; ^3^ DI(s)—dental implant(s).

**Table 5 antibiotics-11-00655-t005:** PICO question 5.

Clinical Problem	PAT ^1^ Prescribing Guidelines for Healthy Patients in the Prosthetic Phase of DIs ^2^ (Second Stages, Impression Taking and Placement of the Implant-Supported Prosthesis).
Population (P)	Healthy patients who are about to start the DI prosthetic phase.
Intervention (I)	Prescription of PAT.
Comparison (C)	No prescription of PAT.
Outcome (O)	Infectious complications.
PICO question	In healthy patients who are about to start the DI prosthetic phase (P), does the prescription of PAT (I), compared with no prescription of PAT (C), decrease the appearance of infectious complications (O)?

^1^ PAT—preventive antibiotic therapy; ^2^ DI(s)—dental implant(s).

**Table 6 antibiotics-11-00655-t006:** PICO question 6.

Clinical Problem	Prescription of Antibiotics Other than Amoxicillin for DI ^1^ Placement in Healthy Patients.
Population (P)	Healthy patients treated with DIs undergoing PAT ^2^.
Intervention (I)	Amoxicillin prescription.
Comparison (C)	Antibiotic different from amoxicillin.
Outcome (O)	DI failure and postoperative infection.
PICO question	In healthy patients treated with DIs and in which PAT is prescribed (P), does the prescription of amoxicillin (I), compared with another type of antibiotic (C), modify the rate of DI failure and/or postoperative infection (O)?

^1^ DI(s)—dental implant(s); ^2^ PAT—preventive antibiotic therapy.

**Table 7 antibiotics-11-00655-t007:** Quick reference summary table of recommendations.

Clinical Situation		Preoperative Regimen	GRADE	Postoperative Regimen	GRADE
Ordinary DIs ^1^ (without anatomical constraints)	NA ^2^	Amoxicillin 2 or 3 g, 1 h before	A	No	-
		No antibiotic prescription	B	No	-
	A ^3^	Azithromycin 500 mg, 1 h before	C	No	-
Immediate DIs (with/without chronic infection of the tooth to be extracted)	NA	Amoxicillin 2 or 3 g, 1 h before	B	500 mg/8 h, 5–7 days	D
	A	Azithromycin 500 mg, 1 h before	D	250 mg/24 h, 5–7 days	D
		Metronidazole 1 g, 1 h before	D	500 mg/6 h, 5–7 days	D
		Clarithromycin 500 mg, 1 h before	D	250 mg/12 h, 5–7 days	D
Sinus lifts (transcrestal and/or lateral window approach)	NA	Amoxicillin/clavulanic acid 875/125 mg/12 h, 1 day before	D	Same regimen, 7 days	D
	A	Ciprofloxacin 500 mg/12 h, 1 day before	D	Same regimen, 9 days	D
Bone Regeneration	NA	Amoxicillin 2 or 3 g, 1 h before	C	No	-
	A	Azithromycin 500 mg, 1 h before	D	No	-
Prosthetic Phase	NA/A	No	D	No	D

^1^ DIs—dental implant(s); ^2^ NA—patients non-allergic to penicillin; ^3^ A—patients allergic to penicillin.

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
