# Peer review of "Consensus Report on Preventive Antibiotic Therapy in Dental Implant Procedures: Summary of Recommendations from the Spanish Society of Implants"

_antibiotics, 2022, doi:10.3390/antibiotics11050655_

Round 1

Reviewer 1 Report

In the era of globalizing bacterial resistance to antibiotics, the subject undertaken by the authors of the article presented to me for review is very up to date. Taking into account the prevalence of procedures related to pre-prosthetic surgery in the area of the oral cavity, the attempt to develop guidelines for  antibiotic prophylaxis fits perfectly into global trends in the fight against antibiotic resistance. An international panel of experts undertook a thorough development of guidelines based on the analysis of the available literature . The formulated guidelines concern not only the indications but also the type of antibiotic and the dosage, taking into account the type of procedure. Such thoroughly developed guidelines may be the starting point for modifying the guidelines of the National Implantology Societies. I strongly recommend accepting the work for publication.

Author Response

Dear reviewer 1,

Thank you for taking the time to review our manuscript and for your kind words.

Best regards.

Reviewer 2 Report

This article is overall well written. Clinical guidelines well presented to surgeons  who performed implants

1. It can be seen as a report that collects opinions of various experts on the use of antibiotics for prophylactic purpose in surgical procedures related to implants in dentistry.

2. The originality of the subject is somewhat lacking. This is because it is a study that relies on expert surveys and is not a biological reserch.

3. Whether prophylactic antibiotics coverage is essential before dental implant placement is a subject of controversy.

   It is common to use antibiotics, mostly depending on the situation. Surgeons are very curious about this. Therefore, what antibiotics

   are used by other surgeons, also in different places, in different countries. This paper is thought to be meaningful because it deals with these points.

4. As the contents is based on a survey, there seems to be a limit for improvement. There is also some subjectivity. The disadvantages is  that the number of participants in this study is small. In addition, it was expressed as an expert dentist in the article, but the selection criteria are ambiguous.

5. I thick it matches

6. References are appropriate

7. Tables and and figures are appropriate

Author Response

Dear reviewer 2,

Thank you for taking the time to review our manuscript. After analyzing your comments, we proceed to answer them one by one.

This article is overall well written. Clinical guidelines well presented to surgeons who performed implants

  1. It can be seen as a report that collects opinions of various experts on the use of antibiotics for prophylactic purpose in surgical procedures related to implants in dentistry.

We appreciate your comment, however, this manuscript is not about recommendations based on the opinions of a panel of experts, but on a panel of experts who conducted a systematic review of the literature for each of the different situations in which implants can be placed (implants on native bone, immediate implants, sinus elevations, bone regeneration, and considerations for the prescription of other types of antibiotics other than amoxicillin). Furthermore, the review carried out was very rigorous and based on the PRISMA criteria.

  1. The originality of the subject is somewhat lacking. This is because it is a study that relies on expert surveys and is not a biological research.

As mentioned in the previous answer, this is a systematic review of the literature. This type of article is the second publication with the highest level of evidence, only preceded by meta-analyses. We only refer in the "introduction" to three surveys to highlight the lack of consensus regarding the prescription of PAT in Oral Implantology and the high number of doses commonly used by professionals when choosing the perioperative guideline as to the most frequent one.

On the other hand, the authors disagree with the lack of originality of our research. Only in the indication "3.2. PAT prescribing guidelines for DI placement in routine situations in healthy patients" various systematic reviews and/or meta-analyses are found. However, in the rest of the indications, not a single one was found. Thus, these recommendations are the first worldwide recommendations on PAT in procedures related to Oral Implantology.

  1. Whether prophylactic antibiotics coverage is essential before dental implant placement is a subject of controversy.

                Yes, it is a controversial issue. What is clear is that in a benefit-risk balance, a prescription made responsibly and based on the present recommendations, can avoid postoperative infections and early failures. Otherwise, these complications may occur, requiring antibiotic guidelines at therapeutic doses, which implies a greater amount of antibiotics and a higher risk of antimicrobial resistance, not to mention possible medical-legal problems that may arise from such complications.

  1. It is common to use antibiotics, mostly depending on the situation. Surgeons are very curious about this. Therefore, what antibiotics are used by other surgeons, also in different places, in different countries. This paper is thought to be meaningful because it deals with these points.

Thank you very much for your comment.

  1. As the contents is based on a survey, there seems to be a limit for improvement. There is also some subjectivity. The disadvantages are that the number of participants in this study is small. In addition, it was expressed as an expert dentist in the article, but the selection criteria are ambiguous.

As pointed out in the response to point 1, the authors did not conduct a survey in our manuscript.

  1. I think it matches
  2. References are appropriate
  3. Tables and and figures are appropriate.

Reviewer 3 Report

I would like to first congratulate the authors for a very good piece of research. A lot of work has been done to be able to put together so much information and to try to summaries it as well as you have.

The authors have done a meta-analysis of studies where antibiotics have or have not been prescribed in dental implant placement surgical steps. The aim was to see if there is a potential benefit in reducing the failure rate and to see where it is advisable not to prescribe antibiotics. The context of this is very well thought of, with the MRSA claiming more and more lives and the rate of dental implant placement and the addressability of this service is raising each year.

The authors have used PICO to split the study intro 6 questions to which they have tried to find and answer for. A quality d assessment was done and a grading to each conclusion/recommendation. 

The research has flawless methodology and reproducibility and is worthy of publication in this present form!

Author Response

Dear reviewer 3,

Thank you for taking the time to review our manuscript and for your kind words.

Best regards.